# Effect of the Hematocrit and Storage Temperature of Dried Blood Samples in the Serological Study of Mumps, Measles and Rubella

**DOI:** 10.3390/diagnostics13030349

**Published:** 2023-01-18

**Authors:** Mariano Rodríguez-Mateos, Javier Jaso, Paula Martínez de Aguirre, Silvia Carlos, Leire Fernández-Ciriza, África Holguín, Gabriel Reina

**Affiliations:** 1Microbiology Department, Clínica Universidad de Navarra, 31008 Pamplona, Spain; 2ISTUN, Institute of Tropical Health, Universidad de Navarra, 31008 Pamplona, Spain; 3Department of Preventive Medicine and Public Health, Universidad de Navarra, 31008 Pamplona, Spain; 4IdiSNA, Navarra Institute for Health Research, 31008 Pamplona, Spain; 5HIV-1 Molecular Epidemiology Laboratory, Microbiology and Parasitology Department, University Hospital Ramón y Cajal-IRYCIS and CIBEREsp-RITIP, 28034 Madrid, Spain

**Keywords:** dried blood spots (DBSs), serology, chemiluminescence, immunoassay, measles, rubella, mumps

## Abstract

Dried blood spots (DBSs) are an economical and convenient alternative to serum/plasma, which allow for the serological and molecular study of different pathogens. Sixty-four blood samples were collected by venipuncture and spotted onto Whatman™ 903 cards to evaluate the utility of DBSs and the effect of the storage temperature for 120 days after sample collection to carry out serological diagnosis. Mumps, measles and rubella IgG were investigated from DBSs and plasma using an automated chemiluminescent immunoassay. Using a calculated optimal cut-off value, the serological evaluation of mumps, measles and rubella using DBSs achieved high sensitivity (100%, 100% and 82.5%, respectively) and specificity (100%, 87.5% and 100%, respectively). The correlation observed between the plasma and the DBSs processed after sample collection was high (0.914–0.953) for all antibodies studied, both considering hematocrit before sample elution or not. For the different storage conditions, the correlation with plasma was high at 4 °C (0.889–0.925) and at −20 °C (0.878–0.951) but lower at room temperature (0.762–0.872). Measles IgG results were more affected than other markers when DBSs were stored at any temperature for 120 days. To summarize, hematocrit does not affect the processing of DBSs in the study of serological markers of mumps, measles and rubella. DBS stability for serological diagnosis of mumps and rubella is adequate when samples are stored at −20 °C or 4 °C, but not at room temperature, for a period of 4 months.

## 1. Introduction

Dried blood spots (DBSs) are a form of sampling in which a few drops of blood are placed on filter paper, which are then allowed to dry at room temperature for several hours and can be easily stored. Ivar Christian Bang is credited for the development of the idea of using blood collected on a paper for analysis when he determined glucose levels from DBSs eluates in 1913 [1]. In 1963, Robert Guthrie began to use DBSs to detect phenylketonuria in newborns using the heel prick test [2]. Since then, several investigators have reported the use of DBSs for serological testing, and it has been used in the surveillance of numerous diseases such as HIV, hepatitis, syphilis and measles [3]. Several studies have demonstrated that antibodies can be detected using DBSs and that the diagnostic accuracy of DBSs is high compared to serum/plasma, indicating that DBSs are a useful alternative to serum [4,5]. In addition, DBSs can also be a reliable sample for molecular testing, as previously reported [3,4,5,6,7,8].

DBSs are an inexpensive, non-invasive and convenient alternative to serum/plasma, easily obtained without the necessary equipment for venipuncture or the qualified healthcare personnel. Furthermore, a low blood volume is required, which can be important in pediatric diagnosis. Its use can be beneficial in low- and middle-income countries where it is difficult to obtain and test serum/plasma samples due to a lack of laboratory resources [9,10]; and also in high-income countries to reach patients not linked to the healthcare system, as those attending Substance Abuse and Mental Health Services, where fingerstick whole-blood samples for DBSs allow the carrying out of serological and molecular diagnosis [7,11,12]. Consequently, this type of sample is well accepted by patients and study participants and may improve the linkage to care [11].

Moreover, the storage and shipment of DBSs are easier than serum/plasma as these cards require less space than a blood tube, are lighter in weight and are not considered infectious material [13]. Due to these advantages, the World Health Organization (WHO) supports the use of DBSs as an alternative to venipuncture [14].

Regardless of the broad use of DBSs for a wide range of serologic studies, there is still not enough evidence for the reliability of DBSs for the analysis of additional biomarkers, such as vaccine-preventable diseases, such as measles, mumps and rubella. This tool, if valid, may facilitate the diagnosis and surveillance of these infectious diseases. Furthermore, differences in the correlation between the serum/plasma and DBSs results have been reported for different pathogens, suggesting that antibodies against certain microorganisms in DBSs may be less stable than others [15]. In addition, there are still no guidelines on how DBSs should be stored to preserve the stability of the different antibodies to be investigated. Therefore, there is a need to guarantee the conditions of DBSs processing and storage to carry out these analyses.

The aim of our study was to evaluate the ability of DBSs to detect serological indicators of three infectious diseases (measles, mumps and rubella) using an indirect chemiluminescent immunoassay (CLIA) and to test the effects of different storage temperatures of DBSs on the stability of these serological markers. In addition, we evaluated the effect of hematocrit on the serological diagnosis of these infections.

## 2. Materials and Methods

### 2.1. Study Design and Participants

A cross-sectional study of 64 adult patients who attended Clínica Universidad de Navarra (Pamplona, Spain) between October and December 2019 was carried out. All of the patients were >18 years old and were selected while taking into account that there was a proportional distribution of men and women, as well as the different age groups and hematocrit levels of the participants. Pregnant women and patients with autoimmune diseases or infectious mononucleosis were excluded due to the higher probability of false positives in the serological analysis. Patients’ names were codified at sampling to maintain confidentiality.

### 2.2. Sample Collection and Storage

For each patient, a blood sample anticoagulated with EDTA was obtained by venipuncture to carry out a complete blood count analysis, including hematocrit value. These samples were collected for routine analysis, and once it was finished, they were selected and used for the research. From this specimen, five DBSs were prepared by spotting 70 μL of whole blood per DBSs on a Whatman™ 903 card (GE Healthcare) that was allowed to dry for 24 h at room temperature. The remaining volume of blood was centrifuged for ten minutes at 2000× *g* to obtain the plasma. After drying, all five DBSs were cut with sterile scissors and inserted into individual Eppendorf microtubes. Of the five spots collected, two dots were processed following sample collection (DBS-A and DBS-B), and the remaining three dots were stored at different temperatures to evaluate stability (at −20 °C (DBS-C), at 4 °C (DBS-D) and at room temperature (DBS-E)) for 4 months before carrying out the analysis.

### 2.3. DBS Elution

Immediately after sample collection (time 0), one dot (DBS-A) was eluted with an adjusted volume of phosphate-buffered saline (PBS), taking into account the hematocrit, to achieve an exact equivalence with the plasma volume used in standard tests (5 µL/assay). The remaining dots were universally eluted (DBS-B, DBS-C, DBS-D and DBS-E) with 1 mL of PBS without considering the hematocrit (Figure 1); all of the dots were incubated at 37 °C for 1 h. To improve elution, the tubes were shaken in a vortex every 15 min. When the elution was finished, the remaining paper was removed from the tube to avoid obstructions in the instrument.

### 2.4. Serological Testing

The detection of IgG against measles, mumps and rubella was performed by indirect chemiluminescent immunoassay (CLIA) using the VirClia^®^ automated system (Vircell), both in DBSs and plasma. Two VirClia^®^ protocols developed by the manufacturer were used, one for plasma with a sample volume of 5 µL and another for eluted DBSs with a volume of 105 µL. The interpretation of the results was carried out following the manufacturer’s instructions considering as gold standard the cut-off values provided for the three pathogens in the plasma/serum, which provides a sensitivity of 96–100% and specificity of 100% according to the manufacturer.

### 2.5. Statistical Analysis

Descriptive statistics were calculated using Excel for Microsoft 365, version 2208. The differences between subpopulations were calculated using a Chi-square test. The assay results for the plasma and DBSs were compared, and correlation was evaluated using Spearman’s test due to the lack of a normal distribution in the samples. To assess the degree of agreement between the two DBS elution methods, the kappa coefficient was calculated. To optimize the use of DBSs for IgG measurements, sensitivity, specificity, positive (PPV) and negative (NPV) predictive values of the tests were calculated for each of the three parameters, considering as gold standard, the results obtained with the plasma. An optimal cut-off index for the interpretation of the DBS results for each parameter was obtained by calculating the area under the curve (AUC) of the receiver operating characteristics (ROC) curves.

## 3. Results

The median age of the 64 participants in the study was 42.6 years, 52% were female, and 90.6% were born in Spain. The general characteristics of the study population can be seen in Table 1.

### 3.1. Immunization Levels in Plasma

Protective IgG levels varied for each disease in the study cohort. We found that protection against measles, mumps and rubella was high, being >75% in all cases (Table 1). The protection coverage rate was higher in women than in men, but the differences were not statistically significant (*p* > 0.05). The percentage of patients with indeterminate plasma results was low (7.8% measles, 1.6% mumps, and 3.1% rubella).

### 3.2. Correlation between DBS and Plasma

The results obtained with the DBS samples showed a strong correlation with those from the plasma samples (Figure 2, Table 2). However, the quantitative results in the immunoassay using DBS samples were slightly higher than those of plasma samples, leading to decreased specificity if the interpretation of the DBS results was performed using the manufacturer’s cut-off values for plasma (1.1). When this cut-off was applied to the interpretation of the DBS results, the percentage of immunized patients for all pathogens was higher than when the plasma results were analyzed, and the specificity was reduced to 60% for measles, 88.5% for mumps and 60% for rubella. Therefore, a new optimized DBS cut-off was calculated for each target using ROC curves to achieve optimal sensitivity and/or specificity in the detection of IgG using DBSs. These values were 1.569, 2.791 and 1.450 for the detection of IgG against mumps, measles and rubella, respectively (Table 3, Figure 3). When the optimized DBS cut-off was applied, the percentage of immunized patients was lower than using plasma for measles (−6.9%), mumps (−1.8%) and rubella (−23%). The use of this new DBS cut-off increased the specificity, PPV and NPV of the tests (Table 3).

#### 3.2.1. Effect of the Hematocrit

When the DBSs were eluted with an adjusted volume of PBS, considering the hematocrit value to analyze exactly the equivalent volume of plasma in standard tests (DBS-A), the results were similar to those obtained when the DBSs were universally eluted with 1 mL of PBS for the three serological markers evaluated (DBS-B) (Figure 2). Both methods obtained excellent concordance between them, with kappa values of 1.0, 0.924 and 1.0 for measles, mumps and rubella, respectively.

#### 3.2.2. DBS Stability

The serological values obtained from the DBSs stored at room temperature for 4 months (DBS-E) differed from those obtained with the DBSs stored at −20 °C (DBS-C) or 4 °C (DBS-D), compared to the results obtained from plasma processed immediately after sample collection (reference result). In addition, DBS-C and DBS-D yielded results comparable to those obtained with DBSs processed without delay (DBS-A and DBS-B), except for the investigation of IgG against measles, where the storage caused a negative effect at any temperature (Table 2). The 4-month processing delay of DBSs reduced the correlation with plasma in 2.43–7.46% when the DBSs were stored at −20 °C, 1.20–6.28% when the DBSs were stored at 4 °C and 4.91–19.61% when the DBSs were stored at room temperature. This drop in the correlation of the DBS samples processed after storage caused a marked reduction in the specificity of the three serological markers. However, since the serological values obtained from DBSs tend to be higher than those from the plasma, the sensitivity of the assays was not affected, except for the evaluation of measles (Table 3). As mentioned above, measles serology results were more affected than mumps or rubella values when the DBSs were preserved for 4 months, with drops in the correlation with the plasma of 7.46%, 6.28% and 19.61% after storage at −20 °C, 4 °C and room temperature, respectively.

## 4. Discussion

The use of DBSs is becoming more and more popular as it is a convenient and suitable sample to carry out different laboratory tests. This study demonstrated equivalent detection of antibodies against measles, mumps and rubella using DBSs compared to plasma. In addition, a universal procedure of DBSs elution independent of hematocrit data has been able to obtain excellent results for antibody detection. Moreover, we observed that the storage of DBSs up to 4 months at low temperatures was adequate to preserve these samples.

Our results demonstrate that DBS samples are valid to verify prior exposure or immunization against some infections, such as measles, mumps and rubella, using an automated chemiluminescent immunoassay. This procedure may also allow the study of the seroprevalence against these vaccinable-preventable diseases and provide useful information to check whether herd immunity has been achieved within a community or target vaccination campaigns. We also report the optimal cut-off values when DBSs specimens are used in the VirClia^®^ automated platform (Vircell) to obtain high sensitivity and specificity when these IgG antibodies are investigated.

The correlation of the quantitative index results obtained from the plasma and DBSs processed immediately after sample collection was excellent, reaching coefficients of 0.914–0.953. Both DBS-A and DBS-B, blood spot specimens processed considering hematocrit level (DBS-A) or not (DBS-B), showed results comparable to those obtained from the plasma. Therefore, it is demonstrated that DBSs elution for serological diagnosis can be performed with a universal elution volume, not exactly adjusted to the hematocrit, thus facilitating the lab work when the patient’s hematocrit data are unknown. This good linearity of the DBSs matrix and plasma with no effect of physiological hematocrit levels on assay performance has been reported previously to measure antibodies against Epstein–Barr or hepatitis E [16,17]; however, the hematocrit may adversely affect the accuracy of therapeutic drug monitoring results where DBSs are a popular sample [18]. Nevertheless, new devices for DBSs collection have recently been proposed to overcome the heterogeneity and hematocrit issues and allow more efficient quantitation [19].

The procedure used with DBSs to carry out the chemiluminescence tests was slightly different to that used with the plasma, as a different sample volume was used to improve the sensitivity and specificity of the analysis. Then, for DBS testing, 105 microlitres of the eluted DBSs were inoculated in the sample well, while the diagnosis from the plasma sample was conducted using 5 microlitres of a sample with 100 microlitres of diluent (1/21 dilution). The optimal cut-off values for the evaluation of the chemiluminescence results of each parameter were calculated by constructing ROC curves. These values were higher than the cut-off index applied to plasma samples (1.1) and allowed for the improvement of the interpretation of the results obtained from the DBSs when compared to those from the plasma. In this way, the procedure for the serological evaluation of measles, mumps and rubella using DBSs achieved high sensitivity (100%, 81% and 100%) and specificity (88%, 100% and 100%), respectively, as previously reported to study infants samples from resource-limited settings [20].

To assess the stability of the DBS samples for serological analysis after a long storage (120 days), three different conditions were evaluated (−20 °C, 4 °C and room temperature). For the different storage conditions, the correlation with the plasma results was high at 4 °C (0.888–0.925) and at −20 °C (0.878–0.951) but lower at room temperature (0.762–0.882). Hence, the storage of DBS samples at room temperature may be suboptimal to carry out serological analysis four months after sample collection, but a shorter period (15–20 days) has not been evaluated and should be validated in the future. Therefore, the long-term storage of DBSs intended for subsequent testing should be undertaken at low temperatures, as previously reported for anti-HIV, HBsAg, anti-HBc, anti-HCV or anti-HEV [21,22,23], as a marked loss of Western blot positivity and low titer antibody signals have been observed if cold storage is not carried out [24].

The negative effect on DBS preservation of storage above 4 °C was observed for the three different antibodies investigated, but, in particular, the values of measles serology were more affected than the other markers. Then, for measles investigation, the diagnosis can be made using DBS since the correlation obtained immediately after collection was high (0.948), but it should be made without delay. The effect of storage may be variable for different biomarkers, as has been observed with different types of antibodies or drug monitoring [25,26,27].

The validation of a commercially available chemiluminescence assay using DBSs for the detection of mumps, measles, and rubella IgG may facilitate the investigation of these markers in low- and middle-income countries where nursing facilities or equipment are not available for sample collection. In addition, the availability of DBSs in high-income countries may be very convenient as a minimally invasive sample, allowing for self-sampling and direct shipment to a clinical laboratory using regular mail. This procedure allows for the investigation of these antibodies, either to study the population that does not regularly attend healthcare services or those in which sample collection may be challenging. Surveillance studies to verify protection against these pathogens are regularly conducted in different situations, especially in occupational safety and health services, and also during pregnancy, after exposure, or previous to international travelling.

Multiple determinants have been identified for lower vaccination uptake among migrants for routine and COVID-19 targets. A tailored, culture-sensitive and evidence-informed strategy has been suggested to strengthen vaccination programmes in high-income countries [28]. The collection of DBSs may enable testing vulnerable people to propose catch-up vaccination campaigns, particularly among populations at greater risk, such as migrants or those born years before universal vaccination against these three pathogens was implemented, mainly between 1960 and 1980 in most high-income countries [29]. However, several authors have highlighted the need for additional validation studies of these techniques to carry out serological surveillance from DBSs as a lack of standardization has been observed in the collection, storage and testing methods [30].

Our study is subject to a number of potential limitations. The first is that the vaccination history of participants was unknown, so no information on the expected biomarkers could be obtained. Second, the pre-infection rate was also not available; however, the high vaccination coverage within our population could reassure the possibility of assessing these antibodies. The results are given by subgrouping the data according to age group and hematocrit, although the study was performed after making a homogeneous selection of the participants in terms of age, sex and hematocrit. The study has several notable strengths, including the use of standardized, commercial and validated robust serological platforms to measure values of interest for the study.

In conclusion, this study confirms the validity of DBS samples for the study of serological markers of mumps, measles and rubella. Moreover, the hematocrit does not affect the processing of DBSs to carry out chemiluminescent immunoassays. DBS stability for use in antibody detection against mumps and rubella is adequate when the samples are stored at −20 °C or 4 °C, but not at room temperature, for a period of 4 months.

## Figures and Tables

**Figure 1 diagnostics-13-00349-f001:**
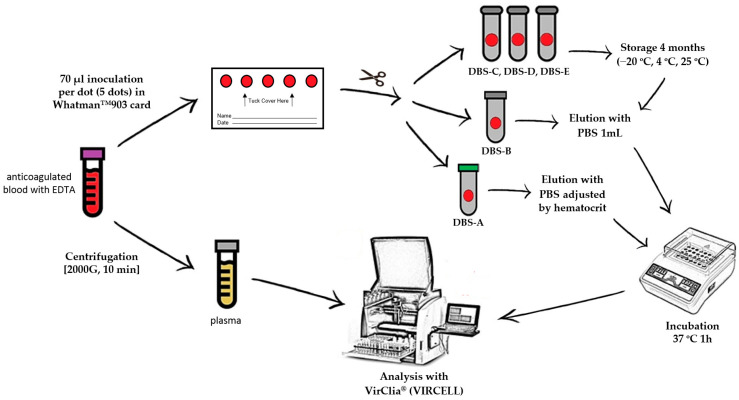
DBS/plasma storage conditions and processing for serological diagnosis. PBS: phosphate-buffered saline; DBS-A: DBSs processed without delay and eluted with an adjusted volume of PBS considering the hematocrit value; DBS-B: DBSs without delay and eluted universally with 1 mL of PBS; DBS-C: DBSs stored at −20 °C for 4 months; DBS-D: DBSs stored at 4 °C for 4 months; DBS-E: DBSs stored at room temperature for 4 months.

**Figure 2 diagnostics-13-00349-f002:**
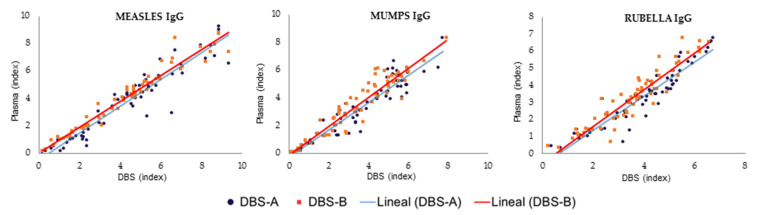
Comparison of DBSs and plasma results for measles, mumps and rubella processed after sample collection. DBS-A: DBSs eluted with a volume of PBS adjusted by hematocrit; DBS-B: DBSs eluted with 1 mL of PBS.

**Figure 3 diagnostics-13-00349-f003:**
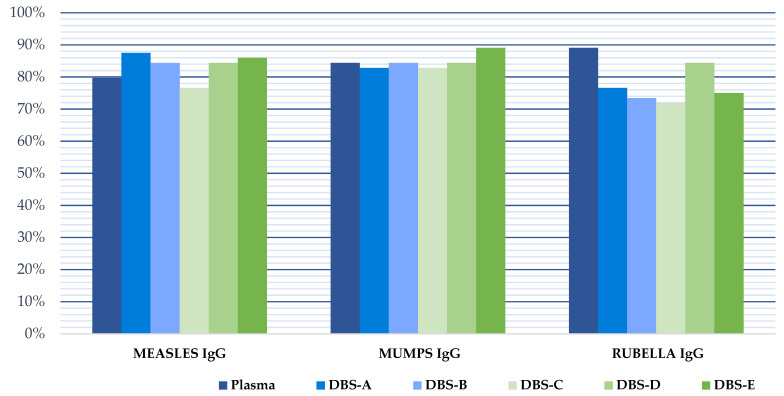
Pathogen immunity against measles, mumps and rubella in our study population considering the manufacturer cut-off for plasma and the optimized DBSs cut-off for all DBSs analysis. DBS-A: DBSs processed without delay and eluted with an adjusted volume of PBS, taking into account the hematocrit; DBS-B: DBSs without delay and eluted with 1 mL of PBS; DBS-C: DBSs stored at −20 °C for 4 months; DBS-D: DBSs stored at 4 °C for 4 months; DBS-E: DBSs stored at room temperature for 4 months.

**Table 1 diagnostics-13-00349-t001:** Characteristics of the study population according to age groups.

	Male	Female	Total
**n (%)**	31 (48%)	33 (52%)	64
**Median age at sampling [IQR]**	42.8 [37.6–58.3]	41.8 [31.6–56.7]	42.6 [36.5–58.1]
**Age Groups, n (%)**			
<30	5 (16%)	7 (21%)	12 (18.8%)
30–45	12 (39%)	12 (36%)	24 (37.5%)
45–60	7 (23%)	8 (24%)	15 (23.4%)
>60	7 (23%)	6 (18%)	13 (20.3%)
**Hematocrit, median [IQR]**	**42.7 [38.9–45.7]**	**39.8 [37.2–41.7]**	**41.0 [37.8–43.6]**
Age groups			
<30	42.7%	40.5%	41.9%
30–45	44.8%	41.0%	42.1%
45–60	43.2%	38.6%	41.0%
>60	38.2%	38.1%	38.2%
**Pathogen immunity (plasma)**			
**Measles IgG (%)**	**71.0%**	**87.9%**	**79.7%**
Age groups			
<30	40.0%	85.7%	66.7%
30–45	58.3%	91.7%	75.0%
45–60	100%	87.5%	93.3%
>60	85.7%	83.3%	84.6%
**Mumps IgG (%)**	**80.6%**	**87.9%**	**84.4%**
Age groups			
<30	100%	85.7%	91.7%
30–45	75.0%	91.7%	83.3%
45–60	71.4%	87.5%	80.0%
>60	85.7%	83.3%	84.6%
**Rubella IgG (%)**	**87.1%**	**90.9%**	**89.1%**
Age groups			
<30	80.0%	85.7%	83.3%
30–45	91.7%	91.7%	91.7%
45–60	71.4%	100%	86.7%
>60	100%	83.3%	92.3%

IQR: interquartile range; Age in years; Hematocrit in %.

**Table 2 diagnostics-13-00349-t002:** Spearman’s correlation coefficient between plasma and DBSs results.

	Plasma/DBS-A	Plasma/DBS-B	Plasma/DBS-C	Plasma/DBS-D	Plasma/DBS-E
MEASLES	0.939	0.948	0.878	0.889	0.762
MUMPS	0.914	0.928	0.905	0.917	0.882
RUBELLA	0.953	0.940	0.951	0.925	0.872

DBS-A: DBSs processed without delay and eluted with an adjusted volume of PBS, taking into account the hematocrit; DBS-B: DBSs without delay and eluted with 1 mL of PBS; DBS-C: DBSs stored at −20 °C for 4 months; DBS-D: DBSs stored at 4 °C for 4 months; DBS-E: DBSs stored at room temperature for 4 months.

**Table 3 diagnostics-13-00349-t003:** Results of VirClia^®^-IgG test for the detection of protective IgG against measles, mumps and rubella.

		MEASLES *				MUMPS **				RUBELLA ***			
		Sen	Spe	PPV	NPV	Sen	Spe	PPV	NPV	Sen	Spe	PPV	NPV
DBS-A	Plasma cut-off	100%	57.1%	94.4%	100%	100%	87.5%	98.2%	100%	100%	60%	96.6%	100%
	DBS cut-off **	100%	87.5%	98.1%	100%	98.1%	100%	100%	90%	84.2%	80%	98%	30.8%
DBS-B	Plasma cut-off	100%	60%	96.2%	100%	100%	88.9%	98.2%	100%	100%	60%	96.6%	100
	DBS cut-off	100%	87.5%	98.1%	100%	100%	100%	100%	100%	82.5%	100%	100%	33.3%
DBS-C	Plasma cut-off	96.1%	50%	94.2%	60%	100%	87.5%	98.2%	100%	98.2%	60%	96.6%	75%
	DBS cut-off	92.2%	87.5%	97.9%	63.3%	98.1%	100%	100%	90%	80.7%	100%	100%	31.3%
DBS-D	Plasma cut-off	98%	42.9%	92.6%	75%	100%	62.5%	94.7%	100%	100%	60%	96.6%	100%
	DBS cut-off	94.1%	62.5%	94.1%	62.5%	98.1%	88.9%	98.1%	88.9%	84.2%	100%	100%	35.7%
DBS-E	Plasma cut-off	96%	0%	88.9%	0%	100%	28.6%	91.5%	100%	100%	40%	95%	100%
	DBS cut-off	94.1%	50%	92.3%	57.1%	98.1%	55.6%	93%	83.3%	80.7%	80%	97.9%	26.7%

* Measles cut-off, plasma: 1.1; DBS: 1.569; ** Mumps cut-off, plasma: 1.1; DBS: 2.791; *** Rubella cut-off, plasma: 1.1; DBS: 1.450. Sen: Sensitivity; Spe: Specificity; PPV: Positive predictive value; NPV: Negative predictive value. DBS-A: DBSs processed without delay and eluted with an adjusted volume of PBS, taking into account the hematocrit; DBS-B: DBSs without delay and eluted with 1 mL of PBS; DBS-C: DBSs stored at −20 °C for 4 months; DBS-D: DBSs stored at 4 °C for 4 months; DBS-E: DBSs stored at room temperature for 4 months.

## Data Availability

The data presented in this study are openly available in the Harvard Dataverse https://dataverse.harvard.edu/dataset.xhtml?persistentId=doi:10.7910/DVN/PGJKA9 (accessed on 15 December 2022).

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
