# Peer review of "Effect of the Hematocrit and Storage Temperature of Dried Blood Samples in the Serological Study of Mumps, Measles and Rubella"

_diagnostics, 2023, doi:10.3390/diagnostics13030349_

Round 1

Reviewer 1 Report

INTRODUCTION
The work includes an extensive bibliographical review in which it shows that the drop of fresh blood (DBS) on filter paper has been used to detect glucose levels as well as serological tests to monitor some infectious diseases.
Studies have also been carried out that show that DBS is capable of detecting antibody tests with results similar to those obtained in serum or plasma.
The DBS has been considered an alternative to detect experiments, it is economical, technically non-invasive with which it obtains results similar to those obtained with serum or plasma. Because precisely a low volume of blood can be used in pediatrics, in patients not linked to the health system. Also in cases of mental illness and drugs abuse.
Another advantage is that it requires little space for storage and is not considered infectious material. Even the OMS supports the use of DBS as an alternative to venipuncture.
Although some serological studies have been carried out, there is insufficient evidence of DBS for the analysis of biomarkers of preventable diseases such as measles, mumps and rubella. However, differences have been found in the comparative data obtained between serum-plasma and DBS for some pathogens. There is also no storage guideline to maintain the stability of the antibody.
The objectives of the work are to evaluate the ability of DBS to detect serological indicators of measles, paper and rubella by indirect immuno-chemiluminescence (CLIA) and to test whether the effects of different DBS storage temperatures it is modify the stability of these serological markers.
In addition, it is evaluated whether the change in hematocrit modifies the serological diagnosis of these infections.

MATERIAL AND METHODS
 The study design is well defined. We believe that the number of samples chosen is sufficient for this type of work. Confidentiality rules were correctly applied.
  Sample collection and storage, DBS elution, serological tests by indirect immuno-chemiluminescence (CLIA) using a Virclia automated system in serum and plasma are described in detail, as well as the two protocols developed by the manufacturer for serum and plasma. plasma. The statistical treatment of the data has been carried out with great methodological rigor.
RESULTS
It is described that more than 75% of the subjects analyzed were protected against three microorganisms analyzed, with higer proportion  in women. There is a good correlation between the DBS and plasma samples.
Quantitative results of the enzyme immunoassay with DBS samples were superior to those with plasma, which decreases the specificity. To correct this problem, a new DBS limit had to be calculated using the ROC curve to optimize the value of the result.
The elution of the hematocrit does not affect the detection of the serological markers evaluated.
DBS storage for up to 120 days at low temperature was adequate to preserve the samples.
When analyzing the different storage conditions, the correlation with plasma was high at 4 °C and -20 °C but lower at room temperature. When DBS was stored at any temperature for more than 120 days, measles IgG detection was affected.
The results are conclusive in relation to the proposed objectives.

DISCUSSION AND CONCLUSION
This study demonstrates that measles, mumps, and rubella antibody detection is similar  using DBS compared to plasma. This represents an advantage in diagnosis from the point of view of public health since it turned a method to be very practical.
This procedure makes it possible to study seroprevalence, provides information on collective immunity and in vaccination campaigns.
This work also has the value of establishing optimal cut-off points when DBS samples are used on Virclia (Vircell) platforms. Also this study is carried out with great methodological precision.
It has special diagnostic interest in daily practice for the detection of antibodies.
In my opinion this article  it meets all the requirements to be accepted and published.

Author Response

We would like to thank the reviewers and the editor for the time and effort spent evaluating the earlier version of the manuscript. Below you will find our point-by-point responses to the comments.

Extensive edition of English language and style requested by Reviewer 1 has been carried out in the manuscript. All changes are displayed with track changes.

Yours sincerely,

Gabriel Reina

Reviewer 2 Report

The authors evaluate the use of the dried blood spot method, taking into account storage time and hematocrit for the detection of IgG antibodies against three vaccinable viruses (measles, mumps and rubella). The study is well designed, the method appropriate and the results are clearly stated.

There are some aspects that should be clarified or corrected according to the opinion of this reviewer:

Line 22: It is not necessary to specify the method in the abstract and it is enough to define it as a commercial chemiluminescent assay.

Line 73: The objective of the study includes six infectious agents, but only three were analyzed.

Line 85: How were the patients collected? Describe if the study was accepted by an Ethics Committee and if the consent of the patients was requested.

Lines 249-253: The analysis of the ROC curves to calculate the optimal threshold values must be included in the results section and can be commented on in the discussion.

Lines 302-307: As a final conclusion, do you consider that storage at room temperature is not adequate for the determination of any of the tests evaluated? It is a major impediment to the use of the method in low-income countries.

Author Response

I would like to thank the reviewers and the editor for the time and effort spent evaluating the earlier version of the manuscript. Below you will find our point-by-point responses to the comments.

English language and style has been improved. All changes are displayed with track changes.

Yours sincerely,

Gabriel Reina
